# Multiple sclerosis and abnormal spermatozoa: A bidirectional two-sample mendelian randomization study

Bo Li[1]ⓔ, Lingang Zhang[2]ⓔ, Qi Li[3], Jianhua Zhang[3], Wei Wang[4], Jianwen Quan[3]*

1 Reproductive Medicine Department, Yuncheng Central Hospital affiliated to Shanxi Medical University, Yuncheng, Shanxi, China, 2 Emergency Department, Yuncheng Central Hospital affiliated to Shanxi Medical University, Yuncheng, Shanxi, China, 3 Urology Department, Yuncheng Central Hospital affiliated to Shanxi Medical University, Yuncheng, Shanxi, China, 4 Anaesthesiology Department, Yuncheng Central Hospital affiliated to Shanxi Medical University, Yuncheng, Shanxi, China

ⓔ These authors contributed equally to this work.

* yczxyyqjw@163.com

**Data Availability Statement:** All relevant data are within the manuscript and its Supporting Information files.

## Abstract

### Objective

Multiple sclerosis (MS) is an autoimmune disease of the central nervous system, and previous observational epidemiological studies have suggested an association between MS and male infertility; male infertility due to sperm abnormalities may result from a number of aetiological factors, such as genetics, autoimmune factors, etc., and there are currently no studies to assess whether MS is associated with sperm abnormalities in men. Therefore, we performed a Mendelian randomization (MR) analysis to assess the causal relationship between MS and abnormal spermatozoa.

### Methods

In this study, independent single nucleotide polymorphisms (SNPs) strongly associated with multiple sclerosis (MS) were identified by mining public genome-wide association study repositories and used as instrumental variables to explore causality. The causal effect of MS on sperm abnormalities was systematically assessed using two-sample Mendelian randomization (MR) techniques, and various analytical models such as inverse variance weighting (IVW), MR-Egger and MR-PRESSO were implemented to dissect the association. In addition, a wide range of sensitivity tests, including Cochran's Q test to detect heterogeneity, MR-Egger intercept analysis to assess bias, leave-one-out to test model robustness, and funnel plot analysis to detect potential publication bias, were implemented to ensure the robustness and reliability of the causal inference results.

### Results

There was a significant causal relationship between MS and abnormal sperm (OR 1.090, 95% CI [1.017–1.168], p = 0.014); The accuracy and robustness of the results were confirmed by sensitivity analysis.

**Funding:** This study was supported by the Project Fund of Basic Research Programme of Yuncheng City, Shanxi Province, China (Project No. YCKJ-2023052 to L.Z.)

**Competing interests:** The authors have declared that no competing interests exist.

## Conclusion

Here we show that there appears to be a causal relationship between multiple sclerosis and abnormal spermatozoa. MS as a chronic disease has a higher risk of concomitant sperm abnormalities in its male patients, and reproductive and fertility issues in men with MS should receive special attention from clinicians.

## Introduction

The World Health Organization defines infertility as the inability to conceive after regular and unprotected sexual intercourse for at least 12 months. Infertility is a significant global health issue, impacting an estimated 8–12% of couples of reproductive age [1]. Male factor infertility is considered to play a role in 50% of infertile couples, being the sole contributor in 20% of infertility cases, and co-contributing with female infertility factors in approximately 30% of cases [2]. A comprehensive review has consolidated the evidence implicating male infertility as a harbinger of augmented risks for chronic conditions, comorbidities, cardiovascular disorders, and oncological diseases, thereby positing male infertility as a prospective biomarker predictive of long-term health outcomes and mortality trends [3]. Drawing from a corpus of research, shared etiologies and risk factors underlying male infertility have been postulated and validated, encompassing lifestyle habits such as tobacco use and alcohol intake, pharmaceutical interventions, a history of testicular infections or ongoing pathologies, environmental toxicant exposures, thermal stress to the testes, traumatic incidents affecting the testicles, as well as issues pertaining to ejaculation and erectile function [4].

Multiple sclerosis (MS) is a chronic autoimmune, inflammatory, and neurodegenerative disease that affects the central nervous system (CNS) [5]. It is the most common cause of neurological dysfunction in young adults, typically occurring between the ages of 20 and 40. Females are affected approximately twice as often as males. MS is characterized by immune dysregulation that leads to the infiltration of immune cells into the CNS, triggering demyelination, axonal damage, and neurodegeneration [6]. Disturbances in redox homeostasis are widely recognised to play a central role in the pathophysiological processes of MS. Research shows that MS patient populations consistently exhibit increased oxidative stress and impaired antioxidant defences, whether in the circulatory system, cerebrospinal fluid (CSF) or brain tissue samples obtained at autopsy [7]. In addition, recent research suggests that abnormally elevated serum iron levels may serve as a potential biomarker of cognitive dysfunction in patients with multiple sclerosis [8]. Multiple factors related to MS can impact fertility, including sexual dysfunction, endocrine changes, autoimmune imbalances, and disease-modifying therapies [9]. Studies have shown that women with MS have lower total fertility rates than those without the condition [10], and a higher incidence of infertility diagnoses compared to women without MS [11]. However, there is limited research on the impact of MS on male fertility [12].

As an increasingly utilized analytical method, Mendelian randomization (MR) is considered an ideal tool for optimizing the design of subsequent randomized trials [13]. By using genetic variants associated with the exposure of interest as instrumental variables, MR can circumvent unmeasured confounding in observational studies and explore the causal relationship between potentially modifiable risk factors and health outcomes [14]. Furthermore, genetic variants have been influencing exposure since conception, indicating that MR can evaluate the long-term impact of exposure on the risk of outcomes [15]. The objective of this study is to

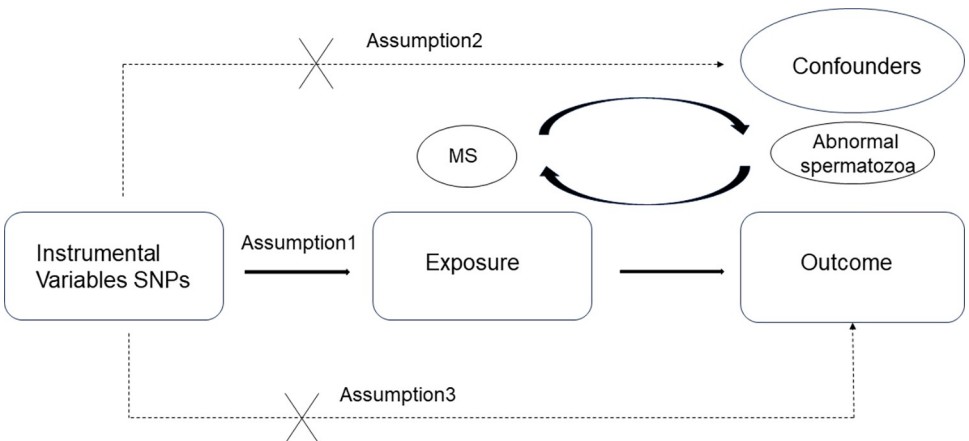

**Fig 1. An overview of the study design.** SNP, single nucleotide polymorphisms. The solid lines represent the association between the instrumental variables (SNPs) and exposure as well as the association between exposure and outcome. Dash lines with means that the association meets two basic assumption of mendelian randomization: i) the genetic variants (SNPs) are independent of confounders between exposure and outcomes; ii) the genetic variants only influence outcome via exposure. MS and Abnormal spermatozoa reciprocal exposure and outcome.

assess the causal effect of MS on abnormal sperm using a two-sample MR analysis and to provide a reference for future research.

## Methods

This is a two-sample Mendelian randomization (MR) study design that employs instrumental variable analysis. The study uses genetic tools, specifically single nucleotide polymorphisms (SNPs), to predict MS and test its causal relationship with abnormal sperm. The genetic variants used in the 2SMR analysis must: a) be closely related to multiple sclerosis, b) be independent of any confounding factors of multiple sclerosis and abnormal sperm, and c) be unrelated to the relevant results obtained by other methods [12] (Fig 1). We conducted a comprehensive search of the exposure and outcome data in the Ieu Open GWAS project database (https://gwas.mrcieu.ac.uk/) to identify the most appropriate GWAS summary data. To mitigate errors resulting from stratification effects, such as ancestry and population, we limited our selection to participants of European descent for the cohort. The analyses were based on publicly available data that have been approved by relevant review boards.

### Data sources

To identify the genetic variants associated with multiple sclerosis (MS), we used the summary data from a publicly available MS GWAS study (47,429 cases and 68,374 controls) [16]. To avoid overlap between the exposure and outcome populations, we used the abnormal spermatozoa GWAS study from the FinnGen biobank, which included 915 cases and 209,006 controls. Both GWAS studies included participants of European ancestry. The IEU Open GWAS database provided the summary data for these two GWAS (MS ID: ieu-b-18; abnormal spermatozoa ID: finn-b-R18_ABNORMAL_SPERMATOZ). For more information on the exposure and outcome datasets (S1 Table).

### The selection of instrumental variables

Instrumental variables (IVs) are used in MR analysis as intermediaries between the exposure and the outcome to explore the causal relationship between them. IVs are typically genetic

variants, with single nucleotide polymorphisms (SNPs) being the most commonly used. The SNPs associated with abnormal spermatozoa were extracted from the IEU Open GWAS project (https://gwas.mrcieu.ac.uk/). First, to ensure the independence of the selected genetic instrumental variables (IVs), we performed a linkage disequilibrium (LD) clustering analysis using 10 megabases (MB) as the clustering window and referring to the European population data, excluding SNPs with high P-values according to the criterion of an LD R2 greater than 0.001 [17]. Second, any SNPs with an extremely significant correlation (P value $<5\times10^{-6}$) with the study phenotype were also discarded to reduce confounding [18]. For SNPs that were not directly included in the target GWAS dataset, we complemented them by searching for suitable surrogate SNPs with an R2 greater than 0.8; if no sufficiently suitable surrogate could be found, the corresponding SNPs were discarded. Immediately following this, the strength of each genetic tool was assessed using the F-statistic, the F statistic was calculated using the following formula: F = β2exposure/SE2exposure. An F statistic $\geq10$ indicates that there is no strong evidence of weak instrument bias [19]. We also excluded intermediate frequency SNPs with allele frequencies greater than 0.42, which are not conducive to subsequent causal inference because the direction of effect is not easily determined. Finally, we applied the MR-PRESSO (Mendelian Randomization Pleiotropy RESidual Sum and Outlier) test to identify and remove SNPs that may harbor pleiotropy, thereby refining the analysis set and increasing the confidence in the results [20].

## Statistical analysis

In order to determine the causal effect of MS on abnormal sperm parameters, the present study used different MR analysis strategies, specifically the Inverse Variance Weighted (IVW) method, the Weighted Median method and the MR-Egger method. The IVW method was employed as the principal analytical strategy for estimating causal effects, given its recognition as the most statistically efficacious approach in discerning causal associations within the context of two-sample Mendelian Randomization (MR) analyses [21]. MR-Egger and Weighted Median complement IVW estimation by providing more robust effect estimates under relaxed assumptions of multivariate validity, although this may be at the expense of statistical power. The weighted median estimator accommodates up to 50% of the instrumental variables (IVs) being invalid or weakly correlated, whereas the MR-Egger methodology tolerates potential invalidity across all IVs. Consequently, heightened confidence in the findings is attained when concordance among these three methodologies is observed, reinforcing the robustness and credibility of the inferred causal relationships. If we observe inconsistent estimation results between different MR methods, we will adopt a more stringent strategy of setting thresholds for instrumental variable p-values to further increase the robustness of the analyses and reduce potential bias [22].

We used the inverse variance weighted (IVW) indicator of heterogeneity (based on the Cochran Q test with a p-value < 0.05) to indicate the possible presence of horizontal pleiotropy. In addition, intercept values obtained from MR-Egger regression analyses were used as an indication of the presence or absence of directional pleiotropy, with p-values less than 0.05 considered evidence of directional pleiotropy [23]. To further analyse and adjust for the effects of horizontal pleiotropy, we used the MR-Pleiotropic Residuals and Outlier Detection (MR-PRESSO) method [22]. This method consists of three key steps: first, identifying horizontal pleiotropy; second, correcting for the effects of pleiotropy by excluding outlier SNPs; and third, comparing causal effect estimates before and after correction to test the significance of the differences. In particular, MR-PRESSO showed lower bias and higher precision than IVW and MR-Egger when the proportion of variance in horizontal polytomousness was less than

10% [20]. We also performed leave-one-out cross-validation to assess the contribution of each single nucleotide polymorphism (SNP) to the overall Mendelian randomization (MR) effect estimate and its potential impact on bias. The MR-PRESSO (Mendelian Randomization Pleiotropy RESidual Sum and Outlier) method was employed to identify any outlying data points within the analysis. Once outliers were detected, they were promptly excised from the dataset. Subsequently, the MR analysis was repeated to ensure the robustness of the results post-outlier removal [24]. All statistical analyses were carried out utilizing the TwoSampleMR package implemented in R software, version 4.2.0.

## Results

Based on the screening criteria of the instrumental SNPs, we obtained 134 abnormal spermatozoa-independent SNPs from the MS GWAS. To assess potential confounding effects and ensure accurate interpretation of genetic variants, this study used an online analysis resource, the Single Nucleotide Polymorphism Annotator (https://snipa.helmholtz-muenchen.de/snipa3/), to perform multiplicity analyses of SNPs. None of these 134 SNPs were associated with potential confounding factors for abnormal spermatozoa, such as smoking, alcohol consumption, drugs, and infections. Furthermore, 124 out of the 134 SNPs were extracted from the GWAS of abnormal spermatozoa. These 124 SNPs were then used as instrumental variables for MS, each with an F statistic > 10 for the exposure association, indicating a low likelihood of weak bias. S1 Table shows the genes corresponding to each instrumental SNP.

MR analysis shows that MS has a causal effect on the risk of abnormal spermatozoa. The specific results of the MR analysis and sensitivity analysis are shown in Table 1, while the corresponding scatterplot and forest plot are shown in Fig 2, both of which show a positive causal association between MS and abnormal spermatozoa. The association between MS and the risk of abnormal spermatozoa was significant (OR = 1.090, 95% CI = 1.018–1.168, P = 0.014) when analysed using the IVW method, suggesting that MS increases the risk of abnormal spermatozoa. Although not statistically significant, the MR-Egger analysis (OR = 1.083, 95% CI = 0.963–1.218, P = 0.184) using the weighted median method (OR = 1.070, 95% CI = 0.959–1.194, P = 0.226) also showed a trend towards MS being associated with an increased risk of abnormal sperm (see Table 1). The Cochran Q-test showed that the p-values of both the MR-Egger (p = 0.385) and IVW (p = 0.383) analyses indicated that there was no heterogeneity problem, which was further confirmed by the fact that there was no significant difference in the intercept values of the MR-Egger regression (intercept = 0.001; p = 0.890). The funnel plot morphology showed good symmetry (Fig 2), and no single SNP was found to have a significant effect on the overall results in the leave-one-out sensitivity analyses where individual SNPs were excluded one at a time (Fig 2). The results of each of these analyses

**Table 1. MR estimates of assessing the causal association between multiple sclerosis (MS) and abnormal spermatozoa.**

| Exposure | Outcome | NO.SNP | MR method | OR | 95%CI | P |
|---|---|---|---|---|---|---|
| | abnormal spermatozoa | | IVW | 1.090 | 1.018–1.168 | 0.014 |
| MS | | 124 | MR Egger | 1.083 | 0.963–1.218 | 0.184 |
| | | | Weighted median | 1.070 | 0.959–1.194 | 0.226 |
| abnormal spermatozoa | | | IVW | 1.010 | 0.941–1.084 | 0.784 |
| | MS | 5 | MR Egger | 1.184 | 0.747–1.877 | 0.525 |
| | | | Weighted median | 0.998 | 0.926–1.076 | 0.962 |

MR: Mendelian randomization, SNP: single-nucleotide polymorphism, OR: odds ratio, CI: confidence interval, IVW: inverse variance weighted.

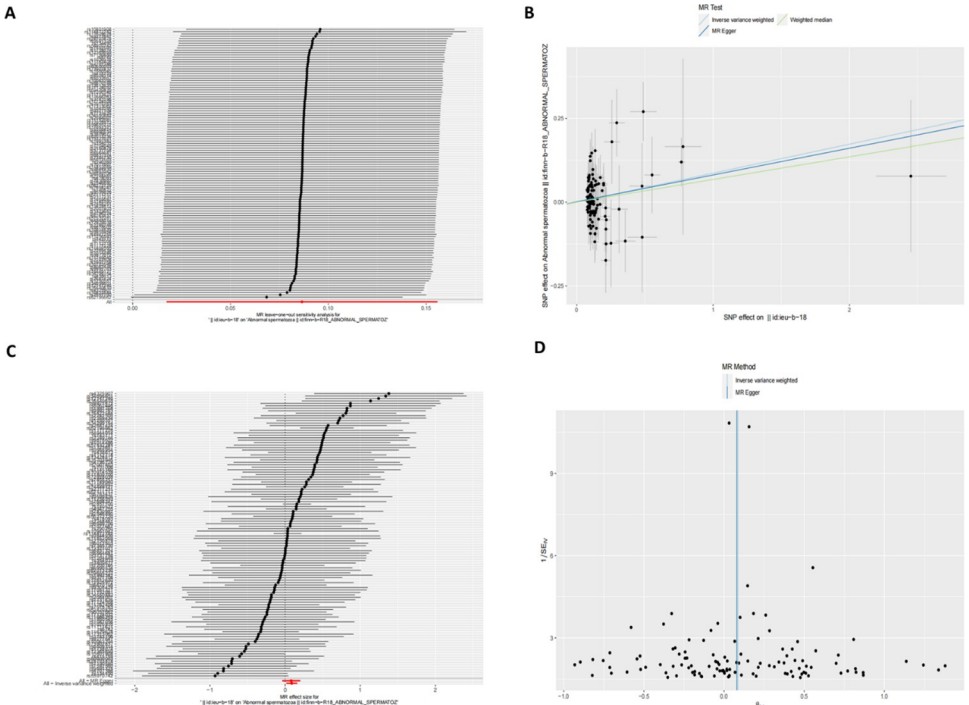

**Fig 2. Mendelian randomization estimates of the associations from MS on abnormal spermatozoa.** Sensitivity analysis (A), scatter plot (B), forest plot (C), and funnel plot (D) of the effect of MS on abnormal spermatozoa.

strongly reinforce the stability and reliability of the conclusions of our Mendelian randomization analysis.

To ensure consistency in the findings across various datasets, we validated our results using the MS GWAS dataset (GWAS ID: ebi-a-GCST90093330) [25]. The analysis revealed a significant association between MS and abnormal spermatozoa (OR 1.128, 95% CI [1.033–1.232], p = 0.007).

To evaluate the reverse causal effect, we used abnormal spermatozoa as the exposure and MS as the outcome. Due to the lack of SNPs with $p<5\times10^{-8}$ in the GWAS summary statistics of abnormal spermatozoa, we selected SNPs with a lower significance threshold ($p<5\times10^{-6}$) [18]. The results indicated no significant association between abnormal spermatozoa and MS (OR 1.010, 95% CI [0.940–1.084], p = 0.784) (S2 Table, Fig 3).

## Discussion

This two-sample MR study based on large-scale genome-wide data demonstrated a consistent effect of genetically predicted MS on increased risk of abnormal spermatozoa. The results remained robust after sensitivity analyses using different Mendelian randomization models.

The epidemiology of MS varies worldwide, suggesting that the aetiology of MS is influenced by a number of geographical and environmental factors [26, 27]. There are geographical variations in the prevalence of MS: the further away from the equator, the lower the sunlight exposure and the higher the prevalence [28]. Numerous studies have shown a direct link between vitamin D deficiency and the risk of MS [29]. As the main source of vitamin D is sunlight-induced synthesis, it is clear that reduced sunlight exposure leads to lower vitamin D levels and thus increased risk of MS [30–32]. Vitamin D plays an important role in innate and acquired immunity as an immunomodulator [33], and also plays a key role in controlling

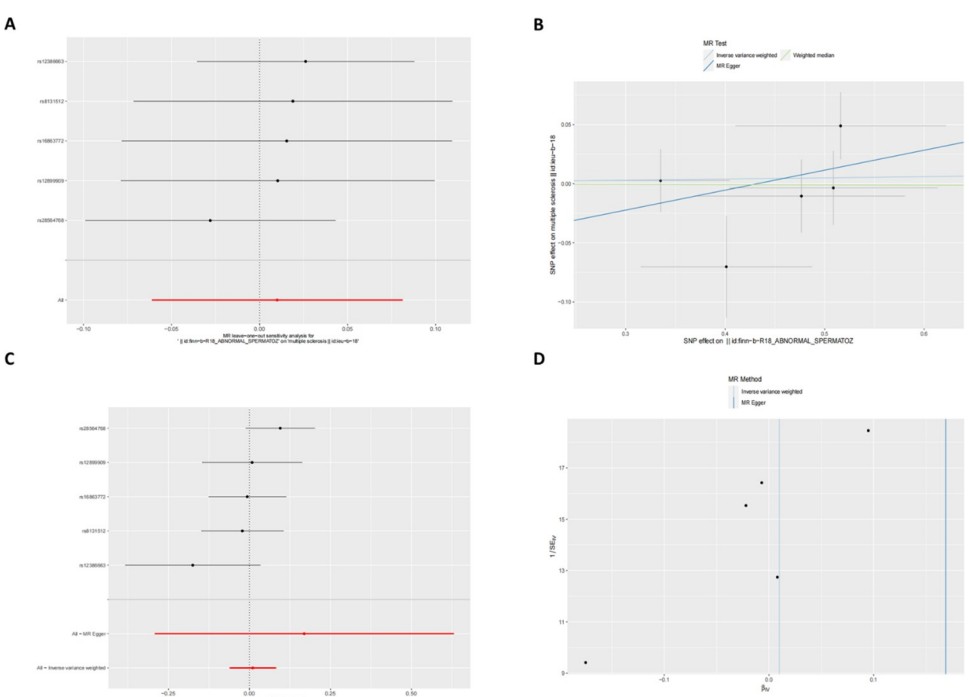

**Fig 3. Mendelian randomization estimates of the associations from abnormal spermatozoa on MS.** Sensitivity analysis (A), scatter plot (B), forest plot (C), and funnel plot (D) of the effect of MS on abnormal spermatozoa.

immune and inflammatory responses in the body [34]. Jensen et al. [35] analysed serum vitamin D levels and sperm quality in 300 male patients and found a positive correlation between vitamin D levels and sperm quality. The authors concluded that vitamin D promotes optimal sperm function. Low vitamin D levels were also associated with a high percentage of D sperm motility, suggesting that low vitamin D levels may be associated with sperm infertility [36]. Some studies [37, 38] have even reported that vitamin D deficiency in vivo may directly or indirectly affect sperm quality, quantity and development, and that impaired sperm function may lead to male infertility. Vitamin D exerts its various biological effects by binding to vitamin D receptors (VDRs) [39], and the expression of VDRs in the testis suggests that vitamin D has potential autocrine and paracrine effects that affect testicular function and possibly male infertility [40].

MS is essentially an autoimmune inflammatory disease of the central nervous system, and the immunopathogenesis of MS may involve the destruction of self-tolerance to myelin and other central nervous system (CNS) antigens, leading to persistent peripheral activation of autoreactive T cells [41, 42]. Upon entry into the CNS, an inflammatory cascade is initiated leading to the release of pro-inflammatory cytokines and chemokines such as interferon-γ, interleukin-2 (IL-2) and tumour necrosis factor (TNF)-α, all of which are key players in mediating inflammation in MS [42, 43], with evidence that TNF-a and interferon-γ receptors are expressed in testicular mesenchymal stromal cells [44, 45] and the hypothalamus [46]. These cytokines can reduce testosterone production through direct action on the interstitial cells of the testis [47] and inhibit the production of gonadotropin-releasing hormone (GnRH) and luteinising hormone (LH) [46]. It has been reported that 24% of men with MS have significantly lower testosterone levels than healthy men of the same age [48], and testosterone levels have a very close relationship with fertility in men [49].

According to an epidemiological study, infertile men have a higher risk of developing rheumatoid arthritis, psoriasis, multiple sclerosis, Graves' disease, and autoimmune thyroiditis compared to men who underwent vasectomy [50]. The association between male infertility and multiple sclerosis is stronger than that between episodic multiple sclerosis and may be related to chronic inflammation [51]. In particular, chronic inflammation has been reported in men with low testosterone levels [52]. Studies have shown that there is a positive correlation between testicular hypoplasia and subsequent MS, and that low testosterone levels are associated with higher levels of disability in patients who have been diagnosed with MS. Testicular hypogonadism (a potential marker of hypogonadism) is a risk factor for MS [10]. Although the mechanism of the association between male infertility and autoimmunity is not clear, androgens may have a protective role in autoimmunity, which may be impaired in hypogonadism [53].

Multiple sclerosis can affect male sexual function. Up to 70% of male MS patients experience erectile dysfunction, and up to 50% show ejaculatory changes. Erectile dysfunction in multiple sclerosis can be attributed to suprasacral, parasympathetic, or peripheral autonomic nerve lesions [54]. Additionally, multiple sclerosis may impair hypothalamic-pituitary function, leading to reduced sex hormone levels due to central nervous system damage. There have been reports of hypogonadotropic hypogonadism, particularly in men with rapid disease progression [55]. This may be one of the reasons why MS affects male fertility.

Reactive oxygen species (ROS) have been reported to be involved in the pathogenesis of a variety of diseases, including several chronic inflammatory diseases [56]. Studies have shown that the pathogenesis of all forms of MS involves inflammation-induced oxidative damage in the CNS, with ROS and reactive nitrogen leading to mitochondrial dysfunction [57]. Other studies have suggested that oxidative stress is more pronounced in the progressive stages of MS and is a key factor in the development of neurodegeneration in patients with progressive MS [58]. While low levels of ROS inhibit sperm capacitation in humans by reducing adenylate cyclase activation, high levels induce sperm lipid peroxidation and DNA damage [59]. Sperm fatty acid membranes contain unstable bonds that are easily oxidized by ROS to produce lipid radicals. These radicals react with nearby fatty acids in a self-perpetuating cycle, forming lipid peroxidation. The degradation of sperm membranes, particularly the midpiece sperm membranes, leads to a decrease in sperm vitality, which is a marker of male infertility [60].

The study used data from the IEU Open GWAS database and the FinnGen biobank to investigate the possible causal relationship between MS and abnormal spermatozoa. Multiple MR methods were employed, with the IVW method demonstrating significantly higher statistical power than other MR methods, particularly the MR-Egger method [61]. IVW was primarily used to screen for potentially significant results. To ensure the robustness of the IVW estimates, sensitivity analyses and other MR methods were performed. If there is horizontal pleiotropy, the IVW estimate may be biased. In such cases, the MR-Egger estimate should be used as it adjusts the IVW analysis by allowing the horizontal pleiotropy effects of all genetic variants to be unbalanced or directional [62]. Consistent beta directions across all MR methods were enforced by most researchers in MR analyses, including this study [63].

Our study found a causal relationship between MS and abnormal spermatozoa, while the opposite was not observed. That is, abnormal spermatozoa did not cause MS. However, our study has some limitations. Firstly, the GWAS summary data we used mainly included Europeans, which may lead to biased estimates and affect generalizability. Secondly, the number of cases of abnormal spermatozoa may not be sufficient, which could introduce bias. The only publicly available GWAS on abnormal spermatozoa did not report specific characteristics such as concentration, vitality, and morphology. As a result, it was not possible to further classify abnormal spermatozoa or perform stratified MR analysis based on specific categories. This

limitation hinders the ability to draw accurate causal inferences and control for potential confounders.

## Conclusions

Studies have found a causal link between multiple sclerosis and abnormal spermatozoa. Men with multiple sclerosis have an increased risk of sperm abnormalities and potential fertility risks. Male reproductive issues are increasingly being studied and discussed, and MS is strongly associated with men's future reproductive health. Clinicians should carry out a thorough reproductive assessment of men with MS, and for male patients with MS who have a need to reproduce, they should intervene early and actively procreate while managing the disease. In addition, the impact of MS on male reproductive function may also be the result of multiple underlying factors, and further research is needed to explore the biological mechanisms underlying this association.

## Supporting information

**S1 Checklist. STROBE-MR checklist of recommended items to address in reports of mendelian randomization studies[1] [2].**
(DOCX)

**S1 Table. Information of the exposures(MS) and outcome(abnormal spermatozoa) datasets.**
(XLSX)

**S2 Table. Information of the exposures(abnormal spermatozoa) and outcome(MS) datasets.**
(XLSX)

## Acknowledgments

We are grateful for all the GWASs making the summary-level data publicly available.

## Author Contributions

**Conceptualization:** Bo Li.

**Data curation:** Bo Li.

**Formal analysis:** Bo Li, Lingang Zhang.

**Funding acquisition:** Jianwen Quan.

**Investigation:** Bo Li, Lingang Zhang, Qi Li, Jianhua Zhang, Wei Wang.

**Methodology:** Bo Li, Qi Li, Wei Wang, Jianwen Quan.

**Project administration:** Lingang Zhang, Qi Li, Jianhua Zhang, Wei Wang.

**Resources:** Jianhua Zhang, Jianwen Quan.

**Software:** Bo Li, Qi Li, Jianhua Zhang.

**Supervision:** Lingang Zhang, Qi Li, Wei Wang, Jianwen Quan.

**Validation:** Bo Li, Lingang Zhang, Jianhua Zhang, Wei Wang, Jianwen Quan.

**Visualization:** Bo Li, Lingang Zhang.

**Writing – original draft:** Bo Li, Lingang Zhang.

**Writing – review & editing:** Qi Li, Jianhua Zhang, Jianwen Quan.

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
