## [Decision Letter · Decision Letter 0]

23 Apr 2024

PONE-D-24-10655Multiple Sclerosis and abnormal spermatozoa: A Bidirectional Two-Sample Mendelian randomization studyPLOS ONE

Dear Dr. quan,

Thank you for submitting your manuscript to PLOS ONE. After careful consideration, we feel that it has merit but does not fully meet PLOS ONE’s publication criteria as it currently stands. Therefore, we invite you to submit a revised version of the manuscript that addresses the points raised during the review process.

We look forward to receiving your revised manuscript.

Kind regards,

David Chau

Academic Editor

PLOS ONE

Journal Requirements:

https://www.frontiersin.org/journals/immunology/articles/10.3389/fimmu.2023.1126457/full

https://wjmh.org/DOIx.php?id=10.5534/wjmh.220109

In your revision ensure you cite all your sources (including your own works), and quote or rephrase any duplicated text outside the methods section. Further consideration is dependent on these concerns being addressed.

Reviewers' comments:

Reviewer's Responses to Questions

**Comments to the Author**

1. Is the manuscript technically sound, and do the data support the conclusions?

Reviewer #1: Yes

Reviewer #2: Yes

2. Has the statistical analysis been performed appropriately and rigorously? 

Reviewer #1: Yes

Reviewer #2: Yes

3. Have the authors made all data underlying the findings in their manuscript fully available?

Reviewer #1: Yes

Reviewer #2: Yes

4. Is the manuscript presented in an intelligible fashion and written in standard English?

Reviewer #1: Yes

Reviewer #2: Yes

5. Review Comments to the Author

Reviewer #1: 10 April 2024

The review on the manuscript, titled ‘Multiple Sclerosis and abnormal spermatozoa: A Bidirectional Two-Sample Mendelian randomization study’ by Li B. et al., submitted to Plos One

Manuscript ID: PONE-D-24-10655

To Authors,

The paper "Multiple Sclerosis and abnormal spermatozoa: A Bidirectional Two-Sample Mendelian randomization study" investigates the causal relationship between multiple sclerosis (MS) and abnormal spermatozoa using Mendelian randomization (MR) analysis. The study obtained instrumental variables (IVs) from publicly available genome-wide association study (GWAS) datasets and performed MR analysis using inverse-variance weighted (IVW), weighted median estimator (WME), and MR-Egger regression models. The results indicate a significant causal relationship between MS and abnormal spermatozoa (OR 1.090, 95% CI [1.017-1.168], p = 0.014), while no significant association was found between abnormal sperm and MS (OR 1.010, 95% CI [0.940-1.084], p = 0.784). The study's findings suggest a potential causal relationship between MS and abnormal spermatozoa, which could have implications for understanding the pathophysiology of MS and male fertility.

The main strength of this paper is the application of a Mendelian randomization (MR) study design to investigate the causal relationship between multiple sclerosis (MS) and abnormal sperm. This analytical approach allows for the exploration of potential causal relationships while minimizing the impact of unmeasured confounding factors, which is a significant advantage in observational studies.

In general, I think the idea of this review is really interesting, and the authors’ fascinating observations on this timely topic may be of interest to the readers of Plos One. However, some comments, as well as some crucial evidence that should be included to support the authors’ argumentation, need to be addressed to improve the quality of the article, its adequacy, and its readability prior to its publication in the present form. My overall judgment is to publish this article after the authors have carefully considered my suggestions below, in particular reshaping the parts of the Introduction and Discussion sections.

Please consider the following comments:

- The abstract could be improved in several ways, as it could benefit from more specific information about the MR methods used, such as the inverse-variance weighted (IVW) model, the weighted median estimator (WME) model, and the MR-Egger regression (MER) model. Furthermore, the presentation of the results in the abstract could be more concise and focused on the key findings. For example, the abstract could mention the odds ratio and confidence interval for the causal relationship between MS and abnormal sperm.

- The introduction could benefit from a more detailed discussion of the neural substrates involved in the relationship between MS and abnormal sperm. I believe that this would provide a stronger foundation for the study and help situate it within the broader context of neuroimmunology and male reproductive health [1-4].

- The study mainly included participants of European descent, which may lead to biased estimates and limit the generalizability of the findings. It would be beneficial to include a more diverse population to enhance the external validity of the results.

- The GWAS summary data used did not report specific characteristics of abnormal spermatozoa, such as concentration, vitality, and morphology. This lack of detailed information could impact the accuracy of the results and hinder the ability to control for potential confounders effectively.

- While the study employed Mendelian randomization (MR) analysis, which is a robust method, there could be further exploration of potential confounding factors that were not addressed in the analysis. Additionally, the study could benefit from a more detailed explanation of the selection criteria for instrumental variables (IVs) to ensure the validity of the MR analysis.

- The study found a causal relationship between MS and abnormal spermatozoa, but it is essential to acknowledge the limitations of MR analysis, such as assumptions of no horizontal pleiotropy. Providing a more nuanced discussion on the implications of the findings and potential alternative explanations would strengthen the paper.

- It would be valuable for the authors to suggest avenues for future research, such as investigating the biological mechanisms underlying the association between MS and abnormal spermatozoa in more detail. Additionally, exploring the impact of other factors on male fertility in the context of MS could provide a more comprehensive understanding.

- Please cite more references. An original study like this typically cites more than 60-70 references.

I hope that, after careful revisions, the manuscript can meet the journal’s high standards for publication. I declare no conflict of interest regarding this manuscript.

Best regards,

Reviewer

References:

1. DOI: 10.3390/biomedicines12030574

2. https://doi.org/10.3390/ijms25020864

3. https://doi.org/10.3390/ijms25052724

4. https://doi.org/10.3390/biomedicines12030613

5. https://doi.org/10.3390/biomedicines8100406

Reviewer #2: The paper is interesting and well written. The authors conducted a Mendelian randomization (MR)

analysis to evaluate the causal relationship between multiple sclerosis (MS) and abnormal sperm.Three different models of MR analysis were performed: the inverse-variance weighted (IVW) model, the weighted median estimator (WME) model, and the MR-Egger regression (MER) model. The methodology is coerent and adequate to the endpoints of the study. The statistical analysis is adequate and well defined. The results are clear and discussion is coerent with the results and the endpoints of the study. The paper may be acceptable for publication.Statistical analysis Minor english editing.

6. PLOS authors have the option to publish the peer review history of their article (what does this mean?). If published, this will include your full peer review and any attached files.

Reviewer #1: No

Reviewer #2: **Yes: **Giuseppe Murdaca

---

## [Author Response · Author response to Decision Letter 0]

6 Jun 2024

Reviewer #1:

1. The abstract could be improved in several ways, as it could benefit from more specific information about the MR methods used, such as the inverse-variance weighted (IVW) model, the weighted median estimator (WME) model, and the MR-Egger regression (MER) model. Furthermore, the presentation of the results in the abstract could be more concise and focused on the key findings. For example, the abstract could mention the odds ratio and confidence interval for the causal relationship between MS and abnormal sperm.

Thank you very much for your careful review, which helped us a lot with the article. Following your suggestions, we have revised the methods and conclusions in the abstract section to make the methods more substantial and the conclusions more concise. Please see page 2 of the revised manuscript, lines 28–39, and page 3, lines 40–41.

2. The introduction could benefit from a more detailed discussion of the neural substrates involved in the relationship between MS and abnormal sperm. I believe that this would provide a stronger foundation for the study and help situate it within the broader context of neuroimmunology and male reproductive health [1-4].

Thank you for your constructive comments on our article, we have introduced relevant literature into our article. Please see page 5 of the revised manuscript, lines 85–92.

3. The study mainly included participants of European descent, which may lead to biased estimates and limit the generalizability of the findings. It would be beneficial to include a more diverse population to enhance the external validity of the results.

Thank you for your suggestion, this is indeed a shortcoming of this study for which we apologise. As the majority of data in the publicly available database is of European ethnicity, data of other ethnicities could not be found, and this shortcoming is also described in the discussion section of the article. We will continue to refine the data content of this study if other ethnicities are subsequently found to be available in the database.

4. The GWAS summary data used did not report specific characteristics of abnormal spermatozoa, such as concentration, vitality, and morphology. This lack of detailed information could impact the accuracy of the results and hinder the ability to control for potential confounders effectively.

Thank you for your careful review and we apologise for this, as with the above response, as the current publicly available database does not include specific breakdowns of specific sperm concentration, viability and morphology and we will continue to refine the data content of this study if data on specific sperm parameter breakdowns are subsequently found to be available in the database.

5. While the study employed Mendelian randomization (MR) analysis, which is a robust method, there could be further exploration of potential confounding factors that were not addressed in the analysis. Additionally, the study could benefit from a more detailed explanation of the selection criteria for instrumental variables (IVs) to ensure the validity of the MR analysis.

Thank you for your valuable comments on the article, following your suggestions we have further analyzed confounders in Mendelian randomization studies and explained the choice of instrumental variables in more detail. Please see page 8 of the revised manuscript, lines 151-171 and page 9, lines 174-216.

6. The study found a causal relationship between MS and abnormal spermatozoa, but it is essential to acknowledge the limitations of MR analysis, such as assumptions of no horizontal pleiotropy. Providing a more nuanced discussion on the implications of the findings and potential alternative explanations would strengthen the paper.

We feel great thanks for your professional review work on our article. We discuss the results and possible alternative explanations in more detail. Please see page 12 of the revised manuscript, lines 229-248.

7. Please cite more references. An original study like this typically cites more than 60-70 references.

Thanks to your suggestion, we have added references cited in the discussion section to explore various aspects of the possible causes of abnormal spermatozoa in MS. Please see page 15 of the revised manuscript, lines 285-323 and page 18, lines 374-356.

Reviewer #2:

The paper is interesting and well written. The authors conducted a Mendelian randomization (MR) analysis to evaluate the causal relationship between multiple sclerosis (MS) and abnormal sperm.Three different models of MR analysis were performed: the inverse-variance weighted (IVW) model, the weighted median estimator (WME) model, and the MR-Egger regression (MER) model. The methodology is coerent and adequate to the endpoints of the study. The statistical analysis is adequate and well defined. The results are clear and discussion is coerent with the results and the endpoints of the study. The paper may be acceptable for publication.Statistical analysis Minor english editing.

Thank you for your suggestions, we have revised the article where the expression was inappropriate and once again we are very grateful for your professional review work.

---

## [Decision Letter · Decision Letter 1]

19 Jun 2024

PONE-D-24-10655R1Multiple Sclerosis and abnormal spermatozoa: A Bidirectional Two-Sample Mendelian randomization studyPLOS ONE

Dear Dr. quan,

Thank you for submitting your manuscript to PLOS ONE. After careful consideration, we feel that it has merit but does not fully meet PLOS ONE’s publication criteria as it currently stands. Therefore, we invite you to submit a revised version of the manuscript that addresses the points raised during the review process.

the editorial office has the following further comments:albeit a large number of SNPs analysed, please ensure to use a relatively high genome wide significance threshold ie p < 10^-8 in order to produce an enough significant SNPsto include details on multiple testing correction to justify for an accurate data analysisit is necessary to use literature or a tool such as phenoscanner to validate whether the identified SNPs are associated with any other outcome or exposureinterpretation and conclusions drawn must be supported by a robust methodology in order to fulfill our publication criteria 3 (https://journals.plos.org/plosone/s/criteria-for-publication). 

We look forward to receiving your revised manuscript.

Kind regards,

David Chau

Academic Editor

PLOS ONE

Journal Requirements:

Reviewers' comments:

Reviewer's Responses to Questions

**Comments to the Author**

1. If the authors have adequately addressed your comments raised in a previous round of review and you feel that this manuscript is now acceptable for publication, you may indicate that here to bypass the “Comments to the Author” section, enter your conflict of interest statement in the “Confidential to Editor” section, and submit your "Accept" recommendation.

Reviewer #1: All comments have been addressed

2. Is the manuscript technically sound, and do the data support the conclusions?

Reviewer #1: Yes

3. Has the statistical analysis been performed appropriately and rigorously? 

Reviewer #1: Yes

4. Have the authors made all data underlying the findings in their manuscript fully available?

Reviewer #1: Yes

5. Is the manuscript presented in an intelligible fashion and written in standard English?

Reviewer #1: Yes

6. Review Comments to the Author

Reviewer #1: 18 June 2024

The 2nd review on the manuscript, titled ‘Multiple Sclerosis and abnormal spermatozoa: A Bidirectional Two-Sample Mendelian randomization study’ by Li B. et al., submitted to Plos One

Manuscript ID: PONE-D-24-10655R1

To Authors,

I am pleased that the authors have addressed my previous suggestions. Prior to publication, I respectfully request that the authors consider my comments and revise the manuscript to meet the high standards of the journal.

1. Keywords: Please include six keywords from Medical Subject Headings (MeSH) in the title and the first two sentences of the abstract.

2. Abstract: I recommend that the authors present the background, methods, results, and conclusion in a proportional order within 200 words without subheadings. The general background (one to two sentences), specific background (two to three sentences), and current issue addressed in this study (one sentence) should all be included in the background before moving on to the objectives. In this subsection, I would like the authors to provide background information, a problem statement, and their reasoning for breaking off. The results section ends with a phrase that places this subsection in a broader context. The conclusion should start with a single sentence that summarizes the main message, such as "Here we show." The authors should describe the potential and progress of this study in the field in the first sentence of the conclusion, followed by two to three sentences that provide a broader perspective that any scientist can comprehend.

3. Conclusion: To effectively communicate the manuscript's main message, I recommend dedicating a single paragraph, approximately 150-200 words long, to highlight the authors' extensive and thorough considerations as esteemed experts in their respective fields. This approach would be advantageous because it emphasizes the importance of their efforts to explain the theoretical implications and practical applications of their findings. It is also critical to discuss potential areas for future research, as well as theoretical and methodological aspects that require further development, in order to fully comprehend the significance of this line of research.

I hope that, after careful revisions, the manuscript can meet the journal’s high standards for publication.

I declare no conflict of interest regarding this manuscript.

Best regards,

Reviewer

7. PLOS authors have the option to publish the peer review history of their article (what does this mean?). If published, this will include your full peer review and any attached files.

Reviewer #1: No

---

## [Author Response · Author response to Decision Letter 1]

18 Jul 2024

Response to Reviewers

Dear David Chau and reviewers:

Thank you again for your letter and the reviewers' comments on our manuscript entitled "Multiple Sclerosis and Sperm Abnormalities: A Two-Way Two-Sample Mendelian Randomised Study" (ID: PONE-D-24-10655).These comments help to improve the scientific rigour of our article. Based on your suggestions and requests, we have made corrections to the revised manuscript. All the revised parts are marked in red in the revised manuscript, which we would like to submit for your kind consideration.

Replies to general comments from the editor:

1. Albeit a large number of SNPs analysed, please ensure to use a relatively high genome wide significance threshold ie p < 10^-8 in order to produce an enough significant SNPs.

Thank you very much for your valuable comments. Increasing the genome-wide significance threshold will reduce the number of SNPs, which will affect the analysis of the data. And there have been several similar studies using a genome-wide significance threshold of p < 10^-6 (PMID: 36077729[1], PMID: 33243239[2], PMID: 32186652[3]). In subsequent studies, we will look for more data on multiple sclerosis and use higher thresholds for data analysis to validate our analysis.

1. Garcia-Etxebarria K, Etxart A, Barrero M, Nafria B, Segues Merino NM, Romero-Garmendia I, Franke A, D'Amato M, Bujanda L. Performance of the Use of Genetic Information to Assess the Risk of Colorectal Cancer in the Basque Population. Cancers (Basel). 2022 Aug 29;14(17):4193. doi: 10.3390/cancers14174193 . PMID: 36077729; PMCID: PMC9454881. IF: 4.5 Q1.

2. Kwok MK, Kawachi I, Rehkopf D, Schooling CM. The role of cortisol in ischemic heart disease, ischemic stroke, type 2 diabetes, and cardiovascular disease risk factors: a bi-directional Mendelian randomization study. BMC Med. 2020 Nov 27;18(1):363. doi: 10.1186/s12916-020-01831-3IF: 7.0 Q1 . PMID: 33243239IF: 7.0 Q1 ; PMCID: PMC7694946.IF: 7.0 Q1 .

3. Chen HY, Cairns BJ, Small AM, Burr HA, Ambikkumar A, Martinsson A, Thériault S, Munter HM, Steffen B, Zhang R, Levinson RT, Shaffer CM, Rong J, Sonestedt E, Dufresne L, Ljungberg J, Näslund U, Johansson B, Ranatunga DK, Whitmer RA, Budoff MJ, Nguyen A, Vasan RS, Larson MG, Harris WS, Damrauer SM, Stark KD, Boekholdt SM, Wareham NJ, Pibarot P, Arsenault BJ, Mathieu P, Gudnason V, O'Donnell CJ, Rotter JI, Tsai MY, Post WS, Clarke R, Söderberg S, Bossé Y, Wells QS, Smith JG, Rader DJ, Lathrop M, Engert JC, Thanassoulis G. Association of FADS1/2 Locus Variants and Polyunsaturated Fatty Acids With Aortic Stenosis. JAMA Cardiol. 2020 Jun 1;5(6):694-702. doi: 10.1001/jamacardio.2020.0246. PMID: 32186652; PMCID: PMC7081150.IF： 14.8 Q1

2. To include details on multiple testing correction to justify for an accurate data analysis.

In this study, a variety of MR analysis methods (including Inverse Variance Weighted, MR Egger, Weighted Median) were comprehensively applied, with the Inverse Variance Weighted (IVW)-based method demonstrating outstanding statistical efficacy and significant superiority compared to other MR analysis strategies. To ensure the robustness of the IVW estimation conclusions, we also performed sensitivity analyses complemented by other MR analysis models.

The Inverse Variance Weighted (IVW) heterogeneity indicator (based on Cochran Q test with p-value less than 0.05) is used to indicate the possible presence of horizontal multidirectionality, and in the present study the Cochran Q test showed that there was no heterogeneity in p-values for both MR-Egger (p=0.385) and IVW (p=0.383) analyses. The intercept values obtained from the MR-Egger regression analyses were used as an indicator of the presence of directional multidirectionality, with p-values less than 0.05 considered evidence of directional multidirectionality, and analysis of the data showed that the intercept values for the MR-Egger regression were not significantly different (intercept=0.001; p=0. 890). Using leave-one-out cross-validation to assess the contribution of each single nucleotide polymorphism (SNP) to the overall Mendelian random (MR) effect estimate and its potential impact on bias, no single SNP was found to have a significant impact on the overall results in the sensitivity analyses where individual SNPs were excluded one at a time (Figure 2).

In addition, the present study followed the current practice of most similar studies by maintaining consistency in the direction of effect sizes (beta values) (IVW: 1.090, MR Egger: 1.083, weighted median 1.070) in the application of all MR analysis techniques, a move that increased the reliability of the experimental results. These are described in detail in the data analysis section of the paper.

3. it is necessary to use literature or a tool such as phenoscanner to validate whether the identified SNPs are associated with any other outcome or exposure.

To assess potential confounding effects and ensure accurate interpretation of genetic variants, this study used an online analysis resource, the Single Nucleotide Polymorphism Annotator (https://snipa.helmholtz-muenchen.de/snipa3/), to perform multiplicity analyses of SNPs.

4. interpretation and conclusions drawn must be supported by a robust methodology in order to fulfill our publication criteria 3 (https://journals.plos.org/plosone/s/criteria-for-publication).

This study used Mendelian randomisation to analyse the data. Mendelian randomisation analysis is a genetics-based statistical tool that uses naturally occurring genetic variation to mimic a randomised trial to investigate causal effects between exposures and health outcomes, overcoming the limitations of observational studies. By using genetic variation as an instrumental variable and analysing its effect on exposure and outcome to infer causality, this method mimics the rigour of a randomised controlled trial but avoids the ethical and practical difficulties of doing so in practice.

Sensitivity analyses play a key role in this framework and are used to validate the stability and reliability of the results. This includes detecting heterogeneity in the data, assessing the pleiotropic effects of genetic variants (i.e. the potential for a genetic variant to affect multiple traits), performing analyses that step down individual genetic markers, restricting analyses to the most reliable genetic variants, and checking for publication bias using funnel plots. These steps ensure that results withstand different assumptions and model variations, thereby increasing confidence in causal inferences, and are central to ensuring the scientific validity and effectiveness of Mendelian randomisation analyses.

Responds to the reviewer’s comments:

Reviewer #1:

1.Keywords: Please include six keywords from Medical Subject Headings (MeSH) in the title and the first two sentences of the abstract.

The context in the abstract section has been changed, thank you for your constructive comments. Please see page 2 of the revised manuscript, lines 22–29.

2. Abstract: I recommend that the authors present the background, methods, results, and conclusion in a proportional order within 200 words without subheadings. The general background (one to two sentences), specific background (two to three sentences), and current issue addressed in this study (one sentence) should all be included in the background before moving on to the objectives. In this subsection, I would like the authors to provide background information, a problem statement, and their reasoning for breaking off. The results section ends with a phrase that places this subsection in a broader context. The conclusion should start with a single sentence that summarizes the main message, such as "Here we show." The authors should describe the potential and progress of this study in the field in the first sentence of the conclusion, followed by two to three sentences that provide a broader perspective that any scientist can comprehend.

We'd like to thank you for these valuable comments, which have been carefully edited and summarised. Please see the abstract on page 2, lines 22-49 of the revised manuscript.

3. Conclusion: To effectively communicate the manuscript's main message, I recommend dedicating a single paragraph, approximately 150-200 words long, to highlight the authors' extensive and thorough considerations as esteemed experts in their respective fields. This approach would be advantageous because it emphasizes the importance of their efforts to explain the theoretical implications and practical applications of their findings. It is also critical to discuss potential areas for future research, as well as theoretical and methodological aspects that require further development, in order to fully comprehend the significance of this line of research.

Thank you for your expert advice on our article. The conclusion section has been carefully amended as requested. Please see the conclusions section on lines 383-394 on page 20 of the revised manuscript.

In response to the valuable feedback from the reviewers, we have meticulously revised our manuscript with the aim of aligning it with the esteemed publication standards of "PLOS ONE", a journal renowned for its high prestige and wide readership. We sincerely hope that this refined submission will merit your consideration for publication in your esteemed journal. We sincerely appreciate your continued support and guidance throughout the review process.

Kind regards

Bo Li

Corresponding author :Jianwen Quan

E-mail: yczxyyqjw@163.com(JWQ)

---

## [Decision Letter · Decision Letter 2]

31 Jul 2024

Multiple Sclerosis and abnormal spermatozoa: A Bidirectional Two-Sample Mendelian randomization study

PONE-D-24-10655R2

Dear Dr. quan,

We’re pleased to inform you that your manuscript has been judged scientifically suitable for publication and will be formally accepted for publication once it meets all outstanding technical requirements.

Kind regards,

David Chau

Academic Editor

PLOS ONE

Additional Editor Comments (optional):

Reviewers' comments:

Reviewer's Responses to Questions

**Comments to the Author**

1. If the authors have adequately addressed your comments raised in a previous round of review and you feel that this manuscript is now acceptable for publication, you may indicate that here to bypass the “Comments to the Author” section, enter your conflict of interest statement in the “Confidential to Editor” section, and submit your "Accept" recommendation.

Reviewer #1: All comments have been addressed

2. Is the manuscript technically sound, and do the data support the conclusions?

Reviewer #1: Yes

3. Has the statistical analysis been performed appropriately and rigorously? 

Reviewer #1: Yes

4. Have the authors made all data underlying the findings in their manuscript fully available?

Reviewer #1: Yes

5. Is the manuscript presented in an intelligible fashion and written in standard English?

Reviewer #1: Yes

6. Review Comments to the Author

Reviewer #1: 30 July 2024

The 3rd review on the manuscript, titled ‘Multiple Sclerosis and abnormal spermatozoa: A Bidirectional Two-Sample Mendelian randomization study’ by Li B. et al., submitted to Plos One

Manuscript ID: PONE-D-24-10655R2

To Authors,

I am pleased that the authors have addressed the issues raised in the previous round. Currently, the manuscript is a well-written research paper with informative layouts, which studies the causal relationship between multiple sclerosis and abnormal spermatozoa using Mendelian randomization analysis. I believe the manuscript meets the journal’s high standards for publication. I am looking forward to seeing more papers written by the same authors.

Thank you!

I declare no conflict of interest regarding this manuscript.

Best regards,

Reviewer

7. PLOS authors have the option to publish the peer review history of their article (what does this mean?). If published, this will include your full peer review and any attached files.

Reviewer #1: No

---

## [Editor Report · Acceptance letter]

9 Aug 2024

PONE-D-24-10655R2 

PLOS ONE

Dear Dr. quan, 

I'm pleased to inform you that your manuscript has been deemed suitable for publication in PLOS ONE. Congratulations! Your manuscript is now being handed over to our production team.

Kind regards, 

on behalf of

Dr. David Chau 

Academic Editor

PLOS ONE